# Review of Underwater Sensing Technologies and Applications

**DOI:** 10.3390/s21237849

**Published:** 2021-11-25

**Authors:** Kai Sun, Weicheng Cui, Chi Chen

**Affiliations:** 1Zhejiang University-Westlake University Joint Training, Zhejiang University, Hangzhou 310024, China; sunkai@westlake.edu.cn; 2Key Laboratory of Coastal Environment and Resources of Zhejiang Province (KLCER), School of Engineering, Westlake University, Hangzhou 310024, China; chen.chi.1990@outlook.com; 3Zhejiang Nektron Intelligent Technology Co., Ltd., Hangzhou 310024, China

**Keywords:** ocean sensing, underwater exploration, submersible, sensor technologies

## Abstract

As the ocean development process speeds up, the technical means of ocean exploration are being upgraded. Due to the characteristics of seawater and the complex underwater environment, conventional measurement and sensing methods used for land are difficult to apply in the underwater environment directly. Especially for the seabed topography, it is impossible to carry out long-distance and accurate detection via electromagnetic waves. Therefore, various types of acoustic and even optical sensing devices for underwater applications have come into use. Equipped by submersibles, those underwater sensors can sense underwater wide-range and accurately. Moreover, the development of sensor technology will be modified and optimized according to the needs of ocean exploitation. This paper has made a summary of the ocean sensing technologies applied in some critical underwater scenarios, including geological surveys, navigation and communication, marine environmental parameters, and underwater inspections. In order to contain as many submersible-based sensors as possible, we have to make a trade-off on breadth and depth. In the end, the authors predict the development trend of underwater sensor technology based on the future ocean exploration requirements.

## 1. Introduction

The ocean occupies 71% of the Earth’s total surface area [1]. Since the birth of the Earth, the climates, environments, and ecosystems of the oceans have been inextricably linked to the land. Understanding the oceans will be a great help in studying the Earth’s climate changes, the methods of environmental protection, and the process of biological evolution. In addition, the oceans contain a great number of resources, including hydrocarbons, minerals, hydrothermal vents, and biological resources, which attracts huge interest from human beings [2,3]. The exploration of the ocean can date back a long time ago. The first global scientific voyage was conducted by HMS Challenger in the 19th century, but the exploration methods at that time were simple and even primitive [4]. Currently, with the development of sensing technologies, a variety of sensing methods based on optics, acoustics, and electromagnetics are widely applied for ocean observation and exploration [5]. Acoustic sensing contains various sonar devices for seafloor mapping, submersible navigation, and underwater object. Optical sensing technologies applied to ocean exploration include underwater imaging for objects inspection, spectrophotometry, and fluorophotometry for environmental parameters monitoring. Technologies of electromagnetics are used for underwater metal detection such as mines and mineral resources, and for underwater inspections of cables and pipelines.

Various types of submersible are crucial to vehicles in ocean exploration. It can be classified as human-occupied vehicles (HOV), remotely operated vehicles (ROV), autonomous underwater vehicles (AUV), hybrid ROV and AUV (HROV or ARV), and underwater gliders [6]. In recent years, with the increased demand for ocean exploitation, multifunctional, and intelligent AUVs, which can be used for multiple tasks, is expected for exploration because of the vast area of oceans and the high-cost exploration with ROVs [7]. The development of new type and multifuncitonal AUVs have also put new demands on the intelligence and miniaturization of sensors.

This paper provides an overview of the commonly used sensing technologies for underwater exploration and lists some state-of-the-art products according to application scenarios during ocean exploration. The rest of the paper is organized as follows. Section 2 introduces the sensing technologies for marine geological survey, including seafloor mapping and resources exploration. In Section 3, submersible navigation and communications are described. After that, measurements of part of essential ocean variables (EOVs) for marine environments are explained in Section 4. In Section 5, underwater inspection technologies are reviewed for archaeology, underwater security, cables, and pipelines inspections. Finally, some conclusions are drawn, and the development trend of underwater sensing technologies is predicted.

## 2. Geological Survey

Although ocean observation has been made for hundreds of years, only a limited percentage of the entire seabed has been measured for depth [8]. The lack of seafloor maps has significantly limited the progress of human investigation and understanding of the oceans. With the advancement of sensing technologies and the increase in human activities in the oceans, low-cost and high-resolution ocean observation devices have been developed and equipped on various platforms, such as single-beam sonar, multibeam sonar, sub-bottom profiler, and side-scan sonar. The Nippon Foundation and the General Bathymetric Chart of the Oceans (GEBCO) plan to cooperate globally to map the whole seafloor topography by the end of 2030 [8]. The detailed charted topography of the whole ocean will have a great help on understanding of marine geology, utilizing the marine mineral and renewable energy resources, monitoring the marine geohazards and providing route surveys for underwater cables and pipelines [9,10]. Because of the efficiency and convenience, remote sensing devices have also become a popular research topic in recent years, such as satellite altimetry, bathymetric LiDAR, and satellite-derived bathymetry (SDB) [10]. However, their resolution and accuracy are still far from measurement at close range [11].

### 2.1. Seafloor Mapping

Dated back to about 1000 B.C. in ancient Egypt, sounding poles and ropes with weights were used to measure water depth [12]. Large-scale exploration based on this method was globally practiced in the 1870s, during the HMS Challenger oceanographic expedition [13]. However, such ‘plumb-line’ measurement was gradually replaced by underwater acoustic techniques in the early 20th centuries, due to unexpected errors [10]. As early as the fifteenth century, Leonardo da Vinci discovered the phenomenon of sound transmission in water [14]. It was not until the early twentieth century that it was used to detect the seabed.

#### 2.1.1. Single-Beam Sonar

The earliest sonar technology to be applied to seafloor exploration was single-beam sonar, consisting of piezoelectric crystals or ceramic transducers to generate and receive acoustic signals [15]. The depth of the seafloor is measured by calculating the time difference between the transmitted and received signals. In the 1920s, it was considered to be the beginning of the echo-sounding era when single-beam sonar was used during the search for the Titanic wreck [9,10]. Single-beam sonar is low-cost, small in size, and can be mounted on different observation platforms, even portable, depending on application scenarios. However, single-beam sonar is similar to a flashlight in that it can only illuminate a small area instead of a whole image of the environment. For large areas of geological survey, more efficient devices must be used, such as multibeam sonar or side-scan sonar.

#### 2.1.2. Multibeam Sonar

Multibeam sonar became commercially available in the 1970s [16]. Multibeam, as it means, can transmit a fan of beams simultaneously and receive echo signals to obtain signals over a swath of seafloor. It is more accurate and efficient than single-beam sonar. As Figure 1 illustrated, the state-of-the-art multibeam sonar can have hundreds of beams, and the swathe angle β can achieve between 120 and 150 degrees [10]. Low-frequency no more than 20 kHz sound beams are often used to detect the full ocean depth because they attenuate slowly so that longer distances can be reached. Since beam angle α is determined by the size of the transducer, a large ‘footprint’ will be projected on the seabed when the multibeam sonar ensonifies a deeper depth. A larger ‘footprint’ will lead to a lower resolution. Multibeam sonar equipped on submersible is commonly operating at high frequencies. Although it decays quickly in seawater, submersibles can measure at a close distance. As a result, the formed footprints are small, while higher resolution can be achieved. During the process of seafloor mapping, vessels must reduce their speed. Otherwise, the accuracy of the mapping will be affected. In addition to the high resolution required for data acquisition, post-processing is also critical, including sound speed profile, anomaly data detection, and strip data stitching processing [17]. These technologies can effectively restrain the effects of non-homogeneous seawater, hull fluctuations caused by waves, and environmental noise. Because of weight and power supply, high-resolution multibeam sonar is usually mounted on the survey vessel with many sensors such as attitude sensor and sound speed measurement. The ‘HydroSweep DS’ multibeam sonar from Teledyne Marine is equipped under ships, which work at frequencies from 14 kHz to 16 kHz for seafloor mapping in deep-sea area [18]. In addition to bathymetric information from 10 m to over 11,000 m, it can acquire side scan and backscatter data for seafloor classification [18]. However, it consumes over 35 kW for power supply. Norwegian manufacturer, Kongsberg, has a solution for seafloor mapping named ‘GeoSwath Plus’, which combines swath bathymetry and side-scan sonar. The sensing system is miniaturized to 12 kg with only 40 W power consumption to have a longer mission duration. The GeoSwath Plus is integrated onto the Remus 100 AUV. Up to 200-m depth can be measured below the AUV when operating with 125 kHz frequency [19].

#### 2.1.3. Side-Scan Sonar

Side-scan sonar is composed of two transducers equipped on towfish, ships, or submersibles, as shown in Figure 2a,b. Conventional side-scan sonar transmits sonar signals from both sides by transducers. Usually, rugged, rough, raised seafloor leads to stronger echo while soft, smooth, or depressed seafloor results in weaker echo. The return sounds cast different shadows in the sonar image. Seafloor substrate composition can be qualitatively analyzed based on the strength of the echoes. As shown in Figure 2c, the middle part will leave a blank because the beam patterns of the two side-scan transducers are deployed to optimize range performance without creating interference from one to the other.

The characteristics of single-beam, side-scan and multibeam sonar are summarized and compared in Table 1.

#### 2.1.4. Sub-Bottom Profiler

The sub-bottom profiler is different from the sonar technologies as mentioned above because it can acquire data of sediments and rocks under the seabed, as shown in Figure 3. Its working principle is similar to that of multi-channel reflection seismic. It sends pulses to the seabed using a single channel source [24]. Due to the acoustic impedance difference between submarine lithological strata, reflected echoes will be different. The sound wave reflection and refraction will occur at the interface of different media. The propagation speed vary as well. The thickness of the sediments and rocks can be calculated by the time between sending and receiving the signal. A profile map can be completed by sending a great number of signals. Various frequencies can be chosen to adapt penetration depth and required resolution according to different applied conditions like sediments density and water depth.

### 2.2. Resource Exploration

The extreme dependence of human society on metal minerals and fossil fuels has led to the rapid depletion of land [2,25]. However, the vast ocean is rich in hydrocarbons and mineral resources, which are anticipated to be explored and exploited.

#### 2.2.1. Mineral Resources

Chemical exchanges during crustal activities in the oceans lead to the form of subsea mineral resources [26,27]. Due to the exploitation cost, commercial and research interests are currently focused on the three types of the most potential mineral resources, namely polymetallic nodules, cobalt-rich crusts, and seafloor polymetallic massive sulfides [26,28]. These ores contain iron, nickel, and manganese metal compounds, widely used for wind turbines, electric car batteries, and solar panels. However, with the high mining costs and the environmental problems during mining processes, no commercial mining in the deep ocean has been carried out in open oceans worldwide by now [28].

Many projects are planned to evaluate the distribution and concentration of ocean mineral resources in the deep ocean. Polymetallic nodules could be predominantly on the surface or partly buried on sediment-covered abyssal plains at water depths of approximately 3500 to 6500 m, while most metal-rich cobalt-rich crusts usually occur on the surface of seamounts, ridges, and plateaus at depths of about 800 to 2500 m [26]. In addition, seawater circulating into and out of oceanic crust through hydrothermal vents leads to the formation of seafloor massive sulfide deposits, as Figure 4 shows.

The whole mining process can last a long time and is associated with a lot of factors [29]. Mineral exploration is the first stage of seabed mining. In addition to geological surveys of the seabed, magnetometers and self-potential surveys are used for mineral resources. Because magnetic ores can cause magnetic field and self-potential anomalies [30].

A magnetometer can be designed as boat towed fish and must keep a certain distance from the ship because electromagnetic equipment on ships will produce electromagnetic interference to it. The company Marine Magnetics has designed an AUV towed Explorer magnetometer, which is smaller and more flexible [31]. Furthermore, a startup company Ocean Floor Geophysics (OFG), provides a self-compensation magnetometer (SCM) system based on a real-time compensation algorithm, which can be integrated inside AUVs [32].

As mentioned above, seafloor polymetallic massive sulfides are often found in the vicinity of hydrothermal vents. The self-potential (SP) survey, as a low destructive and high accurate method of seafloor exploration, is used to identify sulfide deposits during initial surveys [30]. It can detect anomalies of naturally occurring electric fields, which could be caused by mineral deposits like massive sulfides [33]. Since the 1970s, many hydrothermal deposits have been found based on the SP method one after another [34,35,36]. Electric field signal generated by mineral ions dispersed around the seafloor hydrothermal sulfide deposit area is fragile. Only a highly sensitive and accurate acquisition and detection system can obtain the signal of seafloor hydrothermal sulfide deposits.

#### 2.2.2. Hydrocarbon

Hydrocarbon is an organic matter consisting of carbon and hydrogen [37].The exploration process of hydrocarbon is complex and expensive. The location of resources needs to be discovered, and the depth, shape, and volume to be measured. The exploration methods can be classified as non-invasive and invasive. Non-invasive measurements, including magnetometry, gravimetry, and seismic measurements, are firstly carried out to locate the deposits, and then drilling is conducted for further analysis [38]. The seismic measurement is the most used method for hydrocarbon exploration. By sending seismic waves to the seafloor and analyzing the intensity and travel time of the return seismic waves, features of sublayers underwater can be characterized [39,40]. The simplest hydrocarbon is methane, which is considered to be a relatively clean fossil fuel. However, methane is also a vital greenhouse gas, which needs to be prevented from massive leaking when exploitation. Chemosynthesis microbes are the primary productivity for methane-based ecosystems. Methane hydrate, produced under high pressure and low temperature in the deep sea, has already attracted the interest of many countries. Two types of methane sensors are widely used for underwater scenarios: sensors based on tuneable diode laser absorption spectroscopy (TDLAS) and METS methane sensors [41,42]. For both methods, the dissolved methane will pass through a semi-permeable membrane into a detection chamber. However, for the TDLAS, specific wavelength lasers (normally IR) will be transmitted into the detection chamber. The light intensity difference between the incident and detected light will be calculated based on Lambert–Beer Law to gain the methane concentration in water [43]. However, the METS methane sensor is not an optical way. The partial pressure, which is related to the dissolved concentration of methane in the detection chamber, will be measured [44].

## 3. Navigation and Communication

Underwater navigation and communication are the key technologies to locate and control submersibles. Due to the high attenuation in seawater, the Global Navigation Satellite System (GNSS) cannot be used underwater [45]. For long-term tasks, submersibles must be equipped with high-precision navigation systems. Inertial/dead reckoning (DR), acoustic and geophysical navigation are introduced for underwater navigation. In addition, different technologies are developed for underwater communications. However, there is still no high bandwidth, long-distance, and low power consumption communication solution until now.

### 3.1. Location and Navigation

Due to the cable links and professional pilots, ROVs are not as flexible as AUVs. It is also because of this, that AUVs are more dependent on navigation and positioning. Compared to ROVs, AUVs are more flexible and with low cost of use. In addition to the battery life, navigation ranges will also limit the underwater activities of AUVs [46,47]. The inertial/dead reckoning(DR), acoustic navigation, and geophysical navigation technologies are introduced in the following sections and summarized in Table 2.

#### 3.1.1. Inertial/Dead Reckoning

Inertial or dead reckoning uses accelerometers, gyroscopes, or other auxiliary equipment to estimate the current state. INS is one of the most commonly used systems. An INS is a fusion of sensors, processors, and other auxiliary units. Inertial measurement unit (IMU) is the basic hardware for INS, which usually contains accelerators and gyroscopes. It uses accelerometers and gyroscopes to measure the acceleration and direction information of the carrier. By integrating the acceleration in different directions, velocity and position information of carrier can be obtained without external references [48]. Due to the state-of-the-art micro-electromechanical system (MEMS) technologies, IMUs are designed to be small in size, have high inaccuracy, low power consumption, and are well suited for submersible applications. Figure 5 shows the structure of the Ellipse 2 Micro manufactured by SBG Systems, which describes connections among IMU, Attitude and Heading Reference System (AHRS), and INS. AHRS has an extended Kalman filter compared with IMU, which provides roll, pitch, heading, and heave [49]. Additionally, INS connects to Global Navigation Satellite System (GNSS) and odometer, but it will not work underwater.

Despite the rapid response of INS, the error accumulates with time. As a result, the accuracy cannot be guaranteed for long-term applications without other auxiliary devices. Another type, ‘SUBLOCUS DVL’, is an inertial navigation system equipped with a Doppler velocity log (DVL) and fiber-optic gyroscope (FOG), which can provide a 0.8 m position accuracy measurement [50]. DVL provides stable speed information and corrects the output parameters of the navigation system. The combined system has higher precision, reliability, and autonomy.

#### 3.1.2. Acoustic Navigation

Acoustic navigation is the most used way for underwater navigation and positioning. According to the deployment of beacons, it can be classified as the long-baseline system (LBL), short baseline system (SBL), and ultra-short baseline system (USBL), as Figure 6 shows. The acoustic method determines the relative positions of underwater vehicles by using deployed baseline transponders as underwater reference points and calculating signals from them. Among them, LBL is highly accurate, even up to centimeter-level [51]. However, the baseline transponders need to be placed on the seafloor, which is more suitable for applications in waters where operations are often required. Baseline transponders of SBL are usually set on the mother ship. The baseline transponder’s distances and mounting method determine the measurement accuracy, which can be below 10cm. Unlike the other two methods, target distance is measured by signal running time, while target direction is determined by the phase difference of the reply signal. Currently, LBL and USBL are wide commercially used, such as Kongsberg’s HiPAP series of acoustic positioning devices. They can switch modes while having both the high accuracy of LBL and the convenience of USBL. Among them, HiPAP-602 can work at a depth of 7000 m and has an accuracy of 0.02 m [52]. Global Multi-Purpose Ocean Acoustic Network has been a hot topic in recent years. Envisioning a Global Multi-Purpose Ocean Acoustic Network can not only do some monitoring tasks in the oceans but also effectively enhance the range of AUV activities underwater, which will promote the development of the oceans [53].

#### 3.1.3. Geophysical Navigation

In addition to INS and acoustic navigation, geophysical navigation is also available. Geophysical navigation enables positioning through sensors that recognize features of the surrounding environment. The sensors that can be utilized include a compass, depth gauge, underwater cameras, and sonar. Thanks to the development of artificial intelligence, the recognition of environmental features can be done by advanced algorithms. Due to the limitation of distance and attenuation in seawater, this approach is commonly used for hovering AUVs because they can get close to the object of interest [55]. In addition, sonar can also be used for imaging and ranging [5]. Sonar-based geophysical navigation relies heavily on the number of sonars and the quality of imaging.

In general, underwater navigation techniques have advantages and disadvantages, while different navigation methods can be chosen for specific tasks. Error accumulates with INS over long-term use, while acoustic navigation has low accuracy and high latency [45]. Acoustics navigations rely strongly on mothership and deployment of beacons. Geophysical navigation is still challenging because of the low-quality vision in seawater. The fusion of multiple navigation sensors is a trend that allows for universal, accurate navigation by drawing on the strengths of multiple parties [57,58].

### 3.2. Underwater Communication

Due to the strong conductivity of seawater, radio frequency (RF) communications are severely attenuated in the ocean. So, it cannot be used for underwater communication. Communication systems are composed of a transmitter, a communication channel, and a receiver. The transmitter can transmit information by modulating the information signal on the carrier signal. Combined with the characteristics of the ocean, the widely used communication methods include fiber-optic communication, underwater acoustic communication, RF communication, and optical visible light communication. Moreover, some other communication methods, such as quantum communication, are also being under research.

#### 3.2.1. Fiber Optic Communication

Fiber-optic communication enables both long-distance communication and high-speed rate transmission at the same time. However, the disadvantage of fiber-optic communication is also apparent when used for underwater communications. It must make a physical connection between transmitters and receivers, which is inconvenient for underwater vehicles. If the cable is too thick, it will significantly influence the maneuverability of the submersible [59].

#### 3.2.2. Underwater Acoustic Communication

Underwater acoustic communication is currently the most common method of communication underwater. The propagation loss of sound in seawater is much smaller than that of electromagnetic waves. It can achieve communication up to several kilometers [59]. Underwater acoustic communication technology converts the text, voice, image, and other information into electrical signals. After that, the encoder digitizes the information and then converts the electrical signal into an acoustic signal through a transducer. The acoustic signal will carry the information through the seawater medium to the receiving end. The other transducer will then convert the acoustic signal into an electrical signal and decode it to get the information.

However, many factors affect the propagation of sound waves in complex seawater, including speed of sound, multipath, and attenuation. The sound speed is influenced by temperature, pressure, and seawater density. For every degree rise of temperature, sound speed is increased by 1.4 m/s. For every 1 km drop in-depth, sound speed will increase by 17 m/s [60]. Underwater acoustic communication could also be subjected to interference from multipath effects. In addition, there are various environmental noises in seawater, such as wave sounds, biological noise, and traveling boats.

Generally, underwater communication methods are developed from radio communications. Commonly used underwater modulation methods include frequency-shift keying (FSK), phase-shift keying (PSK), and orthogonal frequency-division multiplexing (OFDM). Compared with other modulation methods, OFDM has advantages due to its resilience against frequency selective channels with long delay spreads [61,62,63,64]. In addition, although underwater acoustic communication has been relatively mature, it also faces some problems. Firstly, the propagation speed of acoustic waves underwater is five orders of magnitude lower than the speed of light, resulting in a significant delay in information transmission. Secondly, the bandwidth of underwater acoustic communication is limited, resulting in low transmission capacity. Finally, the equipment of underwater acoustic communication is significant and consumes much power. It is because of these drawbacks that people are still researching other ways for underwater communication.

#### 3.2.3. Radio Frequency Communication

Radio-frequency communication is widely used for communication through the Earth’s atmosphere. It is very challenging to realize the radio communications underwater in this way in the atmosphere. The conductivity in seawater is very high, resulting in the limited penetration distance of electromagnetic waves in seawater. The attenuation of electromagnetic waves underwater increases as the frequency rises. At present, underwater radio frequency communication mainly uses three low-frequency bands: very low frequency (VLF), super low frequency (SLF), and extremely low frequency (ELF) [59]. The frequency range of VLF is 3 to 30 kHz. Its transceiver equipment is expensive because ultra-high-power transmitters and antennas are required. Although the signal of VLF is weak, it has poor concealment and is easy to be detected. SLF has a narrow frequency range of 30 to 300 Hz with a low transmission rate, while the frequency range of ELF is 3 to 30 Hz. The attenuation of ELF is smaller than VLF and SLF so that it can propagate long distances [65]. However, data transmission is also inefficient. Currently, only short commands can be transmitted. Figure 7 below shows the relationship between frequency and transmission distance. As the radio frequency rises, the greater the attenuation, resulting in a shorter transmission distance.

#### 3.2.4. Underwater Visible Light Communication

Underwater visible light communication(UVLC) has been widely studied for underwater communications. As shown in Figure 8, the frequency band used for UVLC is 450–550 nm because it attenuates much less than the RF signal in seawater [66]. Dimtley and Sullian found this phenomenon in 1963 [67]. In the 1970s, the United States began experiments in underwater communication using blue-green lasers at 498 nm. UVLC works with high frequency and has a vital information-carrying capacity. It is capable of large capacity and high-speed data transmission and will be little affected by seawater salinity, temperature, electromagnetic, and other factors underwater. However, it is subjected to refraction and scattering of light by various phytoplankton and suspended particulate matter in seawater, resulting in attenuation. Blue-green laser is also easily affected by various marine life underwater because many marine lives can emit lights of the same wavelength. However, UVLC has good directionality, high concealment, and small size of transceiver equipment. Moreover, the highly directional nature of visible light communication also leads to some problems. For example, UVLC requires a high degree of directionality for the installation of optical transmitters and receivers [68]. It is challenging to ensure the reliability of UVLC channels in the complex and changing marine environment [69].

In summary, the above-mentioned four types of underwater communications have their strengths and weaknesses. As shown in Figure 9, their performances based on bandwidth and transmission range are compared. The four kinds of communications will still be used in short-range underwater communications. Because the complex underwater environments have never been possible to establish an airborne-like long-range, high-bandwidth communication channel using small and low power consumption devices. Fiber optics for long-distance communications will still dominate future underwater communication. Base stations connected by fiber optics will be laid in the ocean, and information will be exchanged remotely between submersibles through the base stations.

## 4. Essential Ocean Variables

Human activities in coastal areas are increasing because the exploration and exploitation of the ocean are becoming more and more intensive. Most of the pollution in the ocean comes from human activities, including oil spills, garbage dumping, and domestic and industrial wastewater discharge [72,73]. The ocean has a vast area, so the source of pollution can be anywhere in any country, which makes it challenging to prevent. In addition, once the ocean is polluted, it will spread globally through ocean currents. Ocean has a considerable influence on the global climate and ecosystem. Although the ocean has a robust self-healing ecosystem, it takes a long time to recover. Therefore, it is vital to have overwhelming monitoring of the ocean environmental conditions, significantly how human activities impact the marine ecosystem. The Global Ocean Observing System (GOOS) has listed many ocean research and assessment variables, called essential ocean variables (EOVs) [74]. They are divided into four classes, physics, biochemistry, biology, and ecosystems, and cross-disciplinary. Most commercially used sensors can cover EOVs in physics and biochemistry. This section introduces some marine parameters from the aspect of the environment, including conductivity, temperature, depth, pH, dissolved oxygen (DO), dissolved CO_2_, turbidity, dissolved organic matter, and nutrients.

### 4.1. CTD—Conductivity, Temperature and Depth

CTD is an instrument applied to measure conductivity, temperature, and depth in the ocean, playing an essential role in submersible navigation and environmental monitoring. Among them, the measurement of salinity, namely conductivity, is the most complex. As early as 1901, Knudsen discovered that seawater salinity could be calculated using electrical conductivity, but it was not until the 1950s that people began to put this method into practice [75]. The first multifunctional salinity sensing instrument, salinity-temperature-depth (STD), was developed [76]. Three years later, to fix the fouling problem when used in seawater, an inductive cell was added to STD [77]. In the following decades, digitalization and microprocessor were added into the development of sensing instruments gradually, which solved salinity ’spiking’ issues caused by a mismatch of sensor response time between temperature and conductivity [78]. After that, the practical salinity scale was proposed, which leads to the generalization and standardization of the measurement [79,80]. In the following 30 years, the practical salinity scale of 1978 (PSS-78) [81] has made significant contributions to the research of the oceans. The most classic CTD of Seabird should be the SBE-41 series CTDs, which is designed for the Argo program to profiling 2000 m underwater. Over 15,000 SBE-41 CTDs have been equipped on Argo floats, which validated the stability of these products. In 2010, TEOS-10 was introduced and gradually replaced the practical salinity scale (EOS-80), which has also been accepted by various organizations worldwide [82]. This new standard calculates the properties of seawater by constructing a Gibbs function for seawater [83]. It considers the composition of seawater, its spatial distribution, and the influence of the central material on the density of seawater, which will significantly advance the development of marine science and related interdisciplinary disciplines. The temperature sensor used in CTD is usually a platinum thermistor because of its high accuracy and broader range. The depth measurement is still obtained by converting the pressure measurement. Their advantages and disadvantages of pressure sensors based on piezoresistive, capacitive, and resonant technologies are compared in Table 3.

### 4.2. Turbidity

Turbidity refers to the resistance to the passing light in solution, which includes scattering light by suspended matter and the absorption of light by solute molecules. The turbidity of water is related to the content of suspended substances in water and their size, shape, and refraction coefficient. Currently, turbidity measurement is based on transmission and scattering methods. The transmission method measures the transmitted light through solutions, but it does not apply to low turbidity measurement. Because in low-turbidity solution, it has little resistance to light, leading to measurement difficulties. The scattering method is better for the measurement of low turbidity solutions because, in a high turbidity solution, multiple scattering occurs, and accurate results cannot be obtained. The ISO-7027 standard specifies that for drinking water, the nephelometry method is used [84]. A turbidimetry can be used for the measurement of seawater turbidity.

### 4.3. Dissolved Oxygen

Dissolved oxygen (DO) refers to the molecular state of oxygen dissolved in water [85]. It can get continuous replenishment mainly by the dissolution of oxygen from the air and the photosynthesis of plants in the water. If seawater is polluted by organic matter, oxygen in the water will be seriously consumed. Moreover, when dissolved oxygen is not timely replenished, anaerobic bacteria in the water will get rapid reproduction, which leads to the corruption of organic matter and makes the water body more polluted and smelly. Therefore, the amount of dissolved oxygen in water is an indicator of the self-purification ability of the water. The commonly used methods for measuring DO include the Winkler method, electrochemical method, and optical method [86,87]. Winkler’s method uses chemical reaction for measurement, which is highly accurate but cumbersome and cannot be measured in-situ. The electrochemical method is based on the current generated by the redox reaction at the electrode to determine the measurement, the measurement efficiency is high, but the reaction electrode is accessible to age, which needs regular maintenance and replacement [87]. The optical method overcomes these disadvantages and is more durable but also more expensive.

### 4.4. Dissolved CO_2_

The dissolved CO_2_ in seawater is an essential chain for global carbon cycling. The ocean is thought to be the most significant carbon sink in the world. It is estimated that the ocean absorbs around a quarter of all CO_2_ emissions. Recent evidence even suggests that this figure could be higher [88]. However, the mass emission of CO_2_ also means more CO_2_ will be dissolved in the ocean and cause many ecological problems, such as seawater acidification, coral bleaching, and climate change. Hence, the monitoring of dissolved CO_2_ also draws much attention from scientists.

For the most used commercial dissolved CO_2_ sensor, non-dispersive infrared (NDIR) technology is applied. The dissolved CO_2_ passes through the silicone hydrophobic membrane into a detection chamber. The partial pressure of the CO_2_ in the chamber will reach an equilibrium with the dissolved CO_2_ in ambient water. Narrow-banded NDIR emitted to the gas sample in the chamber with a wavelength around 4.2 μm, due to the high absorption for CO_2_ to 4.2 μm IR. The IR intensity attenuation is proportional to the amount of CO_2_ based on Lambert–Beer Law [89,90].

The NDIR technology is specific for CO_2_, but many kinds of other gases could use the same theory to be detected and measured. IR interacts with the gas, which has a dipole on the molecule [90]. However, the absorption peaks for different gases also differ. As Figure 10 shows, those absorption peaks at a different wavelength in the mid-IR range become the ‘ID’ for those gases [91]. The concentration of the gases can also be obtained according to Lambert–Beer law.

### 4.5. pH

The acidity of seawater can be obtained by measuring the pH value. It indicates the concentration of hydrogen ions in the solution and reflects the changes of the solution chemically. There are two standard methods for measuring seawater pH: the electrode method and the spectrophotometric method. The electrode method is susceptible to two types of drift, the sensor calibration drift and the environmental drift caused by exposure to seawater bio-fouling. However, impurities in indicators dyne can also lead to the error of the spectrophotometric method [93]. Overall, the spectrophotometric method is more accurate, while the electrode method is more suitable for in situ measurements. Marine sensor manufacturer SeaBird uses the electrode method to measure pH. However, Glass electrode and ion-selective field-effect transistor (ISFET) technologies are applied on the two types of products, HydroCAT-EP and SEApHOx, respectively. ISFET technology from Honeywell is used for seawater between 20 and 40 PSU range with high accuracy (0.02 pH), simple maintenance, and calibration [94]. The HydroCAT-EP can work in all natural waters with 0.1 pH accuracy but requires monthly maintenance and calibration [94].

### 4.6. Dissolved Organic Matter

Dissolved organic matter (DOM) is prevalent in water ecosystems, mainly in the form of carbon, nitrogen, and phosphorus [95]. It can be used as a quantitative biomarker to measure the abundance of the local primary productivity. For example, chlorophylls and carotenoids are critical for photosynthetic processes [96]. The concentration level of dissolved aromatic organic compounds can indicate subsea oil leakage and also the pollutants. The underwater fluorometer is commonly used for identifying and measuring DOM, especially aromatic functional groups in water [97]. The exposure of an organic matter (OM) molecule to an external light source will cause its electron configuration changes by absorbing high-energy photons. The electron is promoted from the ground state to an upper excited singlet state. The reverse process, in which excited electron is transitioned to lower level and lower energy photon released, is named luminescence. Two types of luminescence can be observed: fluorescence and phosphorescence. Fluorescence is caused by the direction of electron transition from an excited singlet to a lower energy level. The phosphorescence process additionally involves the electrons transition to a triplet state with their spin changed [98]. The absorption and fluorescence photon wavelengths are specific to different OM molecules [97]. The different inherent optical properties of bulk OM samples can help researchers to identify them.

DOMs that absorb ultraviolet and visible light are called chromophoric or colored DOM. The DOM fraction, which can exhibit fluorescence in the both ultraviolet and visible range, is called fluorescent DOM [97]. According to the Planck-Einstein relation, short-wavelength ultraviolet light is usually used as the excitation source, and according to the energy conservation principle, the fluorescence photon will be in low frequency or red-shifted. As shown in Figure 11, a fluorescence excitation–emission matrix (EEM) can illustrate a substantial amount of information of the composition and structure of OM. Since different OM molecules are specific to the absorption light and emission fluorescence, each EEM is a specific combination of fluorescence intensities over a range of excitation, and emission wavelengths [97].

### 4.7. Nutrients

The amount of organic carbon fixed by phytoplankton is limited and regulated by the availability of inorganic macronutrients like nitrate, nitrite, ammonium, phosphate, and silicic acid [100]. The unusual increase in nutrients in seawater, especially nitrates, ammonium, and phosphates, leads to algal blooms and excessive growth of water plants, which is harmful to marine life and the environment. The primary source of these nutrients is human activity, including run-off from fertilizers used in terrestrial agricultural applications and discharge of wastewater containing detergents. In addition, increased fluctuations of dissolved oxygen levels and decaying organic matter can also result in eutrophication. Based on Lambert–Beer Law, the measurement of both nitrate and phosphate can be performed spectrophotometrically. For nitrate measurements, the concentration of dissolved in seawater can be measured from its ultraviolet (UV) absorption spectrum [101]. This method enables in-situ real-time measurements and has been widely used in various sensors and submersibles for ocean exploration, including AUV and Argo float, as shown in Figure 12. For phosphate measurements, the corresponding chemical reactions need to be performed first and then measured using spectrophotometry. However, real-time measurement is not yet possible. The working principle of ammonium sensors is normally based on the potentiometric sensing technique, which is an electrochemical way [97]. The potentiometric sensor is composed of two electrodes: a working electrode with an ion-selective membrane coated and an inner reference electrode submerged in a liquid electrolyte. The open-circuit potential (OCP) between the working and reference electrodes will be measured by a high impedance voltmeter without current flow. Since only the target ions can pass the membrane on the working electrode, the OCP can reflect the concentration of target ions [102].

In summary, Table 4 lists some of the most measured marine environmental parameters and measurement methods for marine explorations. The related sensors, which have been commercially used and are suitable to be equipped on submersibles for underwater in-situ measurements, are summarized in Table 4. For the exploration of the whole ocean, close international cooperation is required. For example, the Argo program belongs to the Global Climate Observing System (GCOS) and the GOOS, which is concerned about global climate change and its regional impacts [103]. Figure 12 shows one sort of Argo float, which integrated various miniaturized sensors for the measurements of conductivity, temperature, and depth (CTD), pH, nitrate, etc. It contains many observation technologies, including fluoresces and particle backscatter (FLBB), and in-situ ultraviolet spectrophotometer (ISUS), which have contributed much data for ocean observations.

The parameters and sensors mentioned above are able to reflect the fundamental conditions of the ocean environment. These parameters are crucial in the study of marine environments, climates, and ecology. Due to the complexity of the ocean, a limited number of parameters cannot give a complete picture of the condition of the ocean. In addition to the above-mentioned parameters, it is also necessary to pay attention to the effects caused by their historical events in some sea areas, such as nuclear leaks, oil spills, etc., for example, the Japanese Fukushima nuclear power plant leakage and the decision to dump nuclear waste caused global concern and panic [120]. Despite the optimism of environmentalists, the impact of this event on the surrounding seawaters will last a long time. However, monitoring of nuclear radiation is essential when conducting surveys of the surrounding seawaters and seafood products. The typical radioactive fission product nuclides and heavy nuclides include Sr, Nb, Mo, Tc, Ru, Ag, Te, I, Cs, Ba, La, U, Pu, Am, and Cm, which should be focused [121]. The oil spill also has a significant impact on the marine environment, which damages regional ecosystems and the environment on all fronts [122,123]. Spilled oil can encase the skin of animals and even penetrate their fur, poisoning them to die. The toxic gases accompanying oil severely affect air quality and can be mixed into the atmosphere and drift to various areas. In addition, oil pollution lasts for a long time and is difficult to clean up in the short term. Several oil spills have occurred throughout history, the largest of which was the Deepwater Horizon oil spill, which lasted for more than three months. Furthermore, microplastics have also been a hot topic in environmental research. Due to the large number of petrochemicals used by humans, microplastic pollution is widely distributed. It can be found everywhere, even in the Marianas Trench. Although the harm of microplastics to organisms is still inconclusive, microplastics continue to accumulate everywhere and are difficult to decompose [124]. The quantitative analysis of microplastics is still complex, which cannot be carried out in-situ [125].

## 5. Underwater Inspections

Throughout the thousands of years in human history, many cultural heritages and a significant number of wrecks, including planes and ships, have been submerged underwater. It is not easy to search and investigate in such a vast and complex ocean. During the wreck search of flight MH370, although many ships, satellites, and AUVs equipped with advanced sensors were put into action, it has not been found since 2014 [126]. In addition to wrecks and cultural heritages, cables, pipelines, and other equipment are constructed underwater, which must be tracked and inspected regularly. Surveys in shallow waters can be carried out by diver observation and video records. However, diver’s observations are time-consuming and high-risk. The inspections in marine areas are conducted by ROVs, which are equipped with 3D sonar, and LiDAR [127]. As humans increasingly explore the oceans, more targets need to be searched, detected, and maintained. Due to the successful development of computer vision and algorithms is applied in underwater and sonar images to detect objects. Moreover, in maritime security, underwater object detection technology is also used to identify intruders and ocean mines.

### 5.1. Underwater Detection

Usually, cultural heritages and wrecks are well preserved in low oxygen conditions underwater. However, human activities, including fishing nets, marine geohazards, and seabed geological change, can lead to damages to these monuments [128,129]. In order to help archaeologists better investigate and protect historical sites, it is necessary to do the surveys more efficiently. Currently, commonly used underwater archaeological surveys include visual methods and sonar images, which are used for object detection. In addition, in maritime security, underwater object detection technology is also used to identify moving intruders and ocean mines.

Due to the absorptive and scattering characteristic of seawater, there are many challenges for underwater imaging [130]. As shown in Figure 8, absorption of visible light is better than other wavelengths, especially green and blue light, which can travel hundreds of meters before being completely absorbed by water. In addition, light can further attenuate due to the suspended particulate matter, and substances in water [131]. For the vision system, light needs to travel in three different media (water, viewports, air) as shown in Figure 13. Transparent viewport housing is often made from transparent glass or acrylic, made in a domed shape. As Figure 13 shows, the refraction provokes a pin-cushion distortion, which makes that the most significant reconstruction errors appear at the edges of the target [132,133]. Domed shape viewport housing is designed to eliminate the problem, as Figure 13 shows. Furthermore, an image processing algorithm is another way to calibrate the system.

#### 5.1.1. Objects Detection

As mentioned in Section 2.1, the acoustic method uses the principle of acoustic reflection imaging. Multibeam sonar and side-scan sonar can identify and map out the exposed targets on seabed [135]. Multibeam sonar is highly accurate, but because of the equipment size, it is usually ship-borne. Side-scan sonar can be equipped on AUVs, while the resolution of images is low. In addition, due to images obtained by side-scan sonar being distorted, it contains less 3D spatial information. The sub-bottom profiler is capable of scanning several meters under the seabed to find buried objects. However, it is challenging to identify small-sized objects with their low-resolution [136]. In addition, the multipath effect will limit the application of sub-bottom profiler severely.

In addition, some other methods are used for underwater detection, such as ground-penetrating radar (GPR). GPR is usually considered to have dramatic attenuation underwater. However, it is possible to conduct underwater surveys for shallow waters with low conductivity (<0.02 S/m) [137]. GPR was used for an archaeological activity of Lake Shanglinhu to detect over 3.5m underwater, which provided a simple, fast, and high accurate solution [138]. Moreover, the magnetometer is usually designed as a towfish for metal detection due to electromagnetic interference. However, with the advances of data science and algorithm, self-compensation magnetometer (SCM) technology can remove electromagnetic interference from the vehicle itself [139]. SCM magnetometer can be equipped on AUVs and investigate targets at a close distance, enabling more accurate magnetic anomaly data to be obtained and analyzed. For example, multiple sensors are integrated on AUVs for underwater measurements. As shown in Figure 14, a lightweight AUV for underwater archaeology has been designed and tested in several waters [140]. It has an integrated side-scan sonar, magnetometer, and underwater camera. Such a lightweight AUV is faster, more flexible, and more accurate than ship or diver’s surveys because of the better positioning and maneuverability of the AUVs.

#### 5.1.2. Security Issues

In addition to surveys of vehicle wrecks and cultural heritages, underwater objects detection is also used to solve security issues. The terrorist threat to maritime activities contains an underwater threat, including intruder divers and deployed mines [141]. Divers can be detected by a monocular camera on a moving vehicle with the motion compensation through improved Fourier Mellin Invariant (iFMI) registration [142]. Compared with visual methods, sonar is a better choice because it can provide a more comprehensive monitoring coverage with the lowest cost [143]. A wide range of information from the under-detected objects can be obtained with sonar technologies. Combined with algorithms, divers can be detected by their breathing sounds with passive sonar [144]. Nevertheless, its limit range is up to 40 m because the breathing sound is too weak, which could be covered by environmental noise. The comparison of characteristics between acoustic detection and underwater vision is shown in Figure 15. Acoustic detection has higher accuracy and a broader detection range, while underwater vision has a higher working frequency, higher resolution, and lower cost. A multi-sensor fusion approach is also proposed, which collects visual and acoustic signals from low-cost cameras and hydrophones, respectively. That fused data analyzed by machine learning algorithm helps the system more robust and effective [145].

Another threat to underwater safety is due to ocean mines. They have great threats not only on passing vessels but also on the safety of marine lives. It is more difficult than diver detection because of its silence and shape and smaller size. A special method is needed to differentiate mines from seafloor objects such as rocks. Therefore, active acoustics, passive magnetics, active and passive electro-optics have been used for mine detections by now. Due to the high attenuation of electromagnetic radiation, active and passive electro-optics are often applied in airborne platforms. It has a high coverage rate but low resolution. However, airborne synthetic aperture radar (SAR) is good at searching surface mines [146]. Side-scan sonar as a kind of active acoustic sensor is widely used for mines detection. Even though mines in imagery from electro-optic or sonar sensors, as shown in Figure 16, are still hard to be identified. Fortunately, various intelligent algorithms are developed for pattern analysis, whose accuracy can achieve to 90% [147,148,149].

### 5.2. Track and Inspect

By now, more than 8000 km of power cables, 1.3 million km of submarine telecommunication cables, and 3 million km of active pipelines have been laid under the oceans [7,150]. Figure 17a,b shows the deployed submarine communication cables which connect around the world. The percentages of causing cable damage are classified as a percentage in Figure 17c. Anchoring, fishing, and other human activities mainly lead to cable faults, which could have significant disturbances worldwide [151]. It is necessary to inspect cables regularly.

At present, ROVs and towed systems are mainly used to inspect subsea pipelines and cables. Because of the need for professional pilots, support crews, and ships for ROVs, the inspection cost is high [7]. As demand grows, AUVs and HROVs are considered the next generation of the best alternatives for subsea inspection. However, AUVs are still not put into service because of many challenges, including battery limitations, underwater communications, and navigations [54]. In addition, the tracking sensors that can be mounted on the AUVs are also critical. Currently, visual methods, sonar, and magnetometer are used for subsea cable tracking. The visual method locates and tracks the cable by images. Therefore, the quality of images plays an essential role in identification. However, buried cables cannot be detected, while acoustic waves can penetrate seabed for buried cables. In addition, subsea cables can also be tracked by a magnetometer because of the ferromagnetic, and current-conducting materials [154]. Underwater automatic cable tracking and inspection systems are designed with tri-axial magnetometers for power cables, and optical cables [155,156]. For more high reliability and accuracy, combined information from multiple sensors is analyzed.

The leakage of submarine pipelines is severe for the regional marine environment. Possible failures of pipelines come from incorrect installation, internal or external corrosion, anchoring or trawling of ships and marine geohazards in shallow water [157,158]. Commonly used detection technologies include methods of magnetics, electronics, acoustics, and radiography. The magnetic method can detect both internal and external defects but not axially oriented defects, whose result is usually qualitative [159]. The electric method requires steady contact with the pipeline. The data need to be processed, and the results rely on the magnetic permeability of pipeline materials [160]. Sonar can measure the shape and location of pipelines. However, it is susceptible to pipeline vibrations caused by ambient noise. Radiography can be used for most defects and the layers of pipelines, but it is harmful to nearby humans, and fauna [161]. In addition, distributed fiber optic sensors can be set when designing. It can detect a wide range of defects in a long distance. However, it will be disabled when one point is broken. For an optimal result, multiple tools can be used [157].

## 6. Discussion and Summary

This paper has comprehensively reviewed the mainly used underwater sensing technologies in acoustics, optics, electromagnetics, and more, equipped on submersibles. Over the last century, underwater sensing technologies have shared great benefits from technology development. The sensing devices are in fast progress from rope and weight to a spectrometer. In the following years, it will be a boom of the ocean exploration era. From new materials to the design of submersibles to the new fabrication methods such as 3D printing, it will enable more efficient and intelligent exploration of large oceans. The development of underwater sensing technologies should follow the requirements of ocean exploration. Firstly, some new types of autonomous submersibles, such as soft deep-sea bionic fish [162], are hot research topics. It puts new demands on the development of sensors. Traditional sensors need to be redesigned and updated to prepare submersibles by combining new materials and new fabrication methods. Secondly, with the development and gradually using AUVs, existing sensors are expected to be miniaturized and equipped on AUVs. The miniaturized design of sensors is the size that needs to be considered and also the energy consumptions. In addition, different functional sensors should be integrated into the AUVs to be applied in different scenarios. On the other hand, detected information can be confirmed from different aspects. Moreover, underwater infrastructures like docking stations are fundamental for broad ocean explorations so that autonomous submersibles can continuously work underwater and realize timely exchange information. Furthermore, groups of AUVs or schools of robotic fish for ocean exploration will be developed and applied more and more.

## Figures and Tables

**Figure 1 sensors-21-07849-f001:**
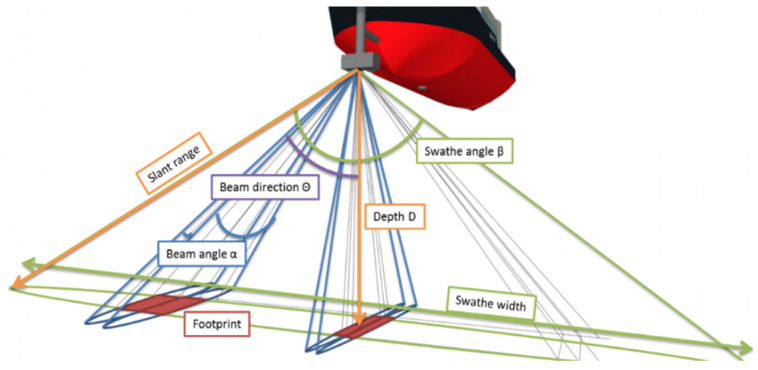
Parameters of a Multi-beam Sonar [20].

**Figure 2 sensors-21-07849-f002:**
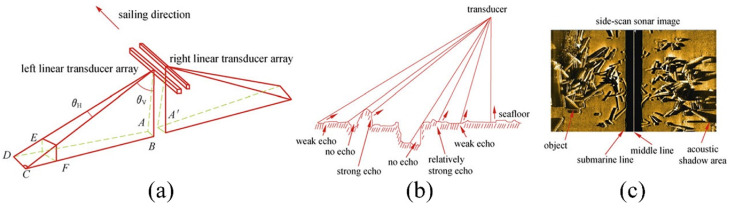
(**a**,**b**) Side-scan sonar working principle. (**c**) Principle of side-scan sonar image generation [21].

**Figure 3 sensors-21-07849-f003:**
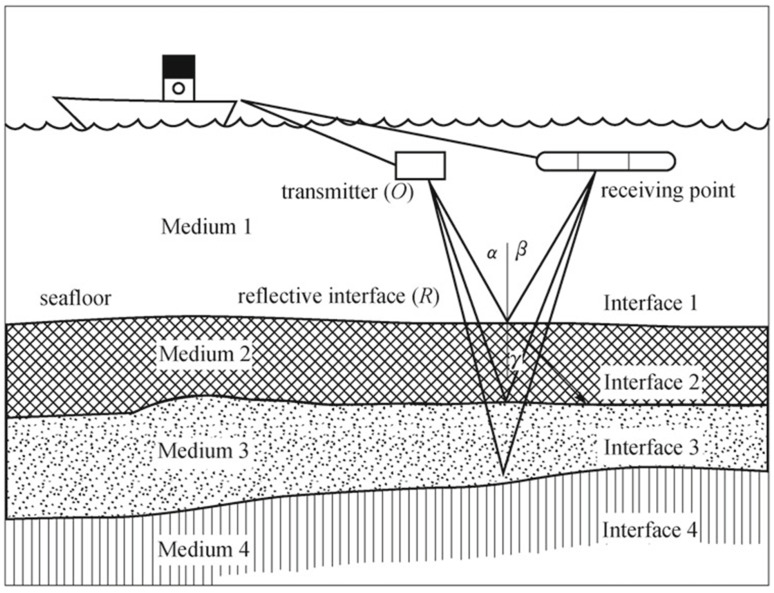
Working principle of sub-bottom profiler detection [21].

**Figure 4 sensors-21-07849-f004:**
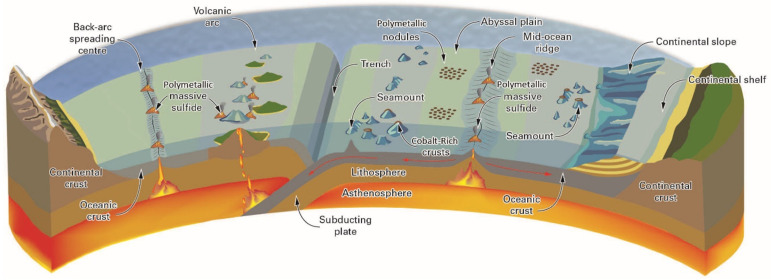
A cross-section through the Earth’s crust and the distribution of the major metal-rich deep-ocean mineral deposits on different types of plate boundary [28].

**Figure 5 sensors-21-07849-f005:**
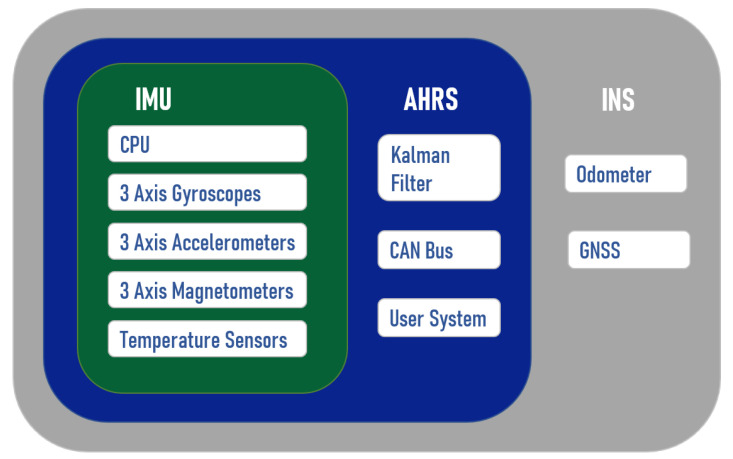
IMU, AHRS, INS and structural diagram of their functions from SBG Systems company, Adapted from [49].

**Figure 6 sensors-21-07849-f006:**
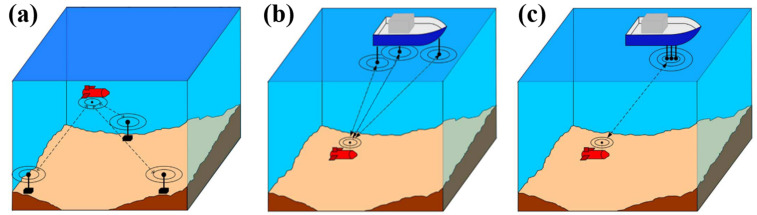
Acoustic Navigation: (**a**) LBL: long baseline system. (**b**) SBL: short baseline system. (**c**) USBL: ultra-short baseline system [54].

**Figure 7 sensors-21-07849-f007:**
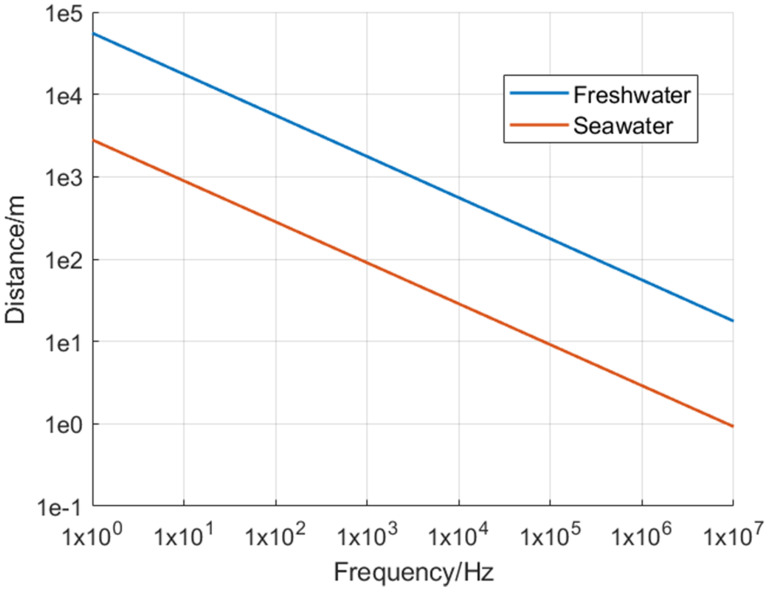
Impact of increasing frequency on propagation distance [65].

**Figure 8 sensors-21-07849-f008:**
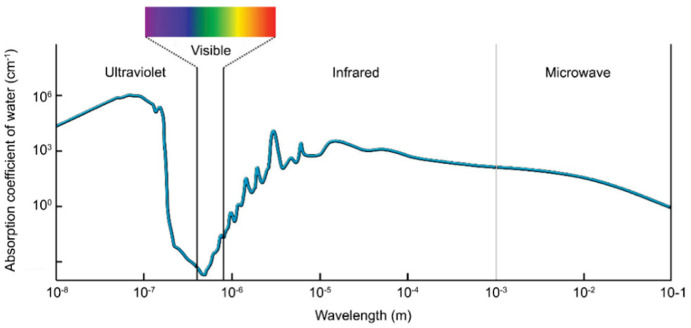
Absorption coefficient of pure seawater for different transmission wavelengths [70].

**Figure 9 sensors-21-07849-f009:**
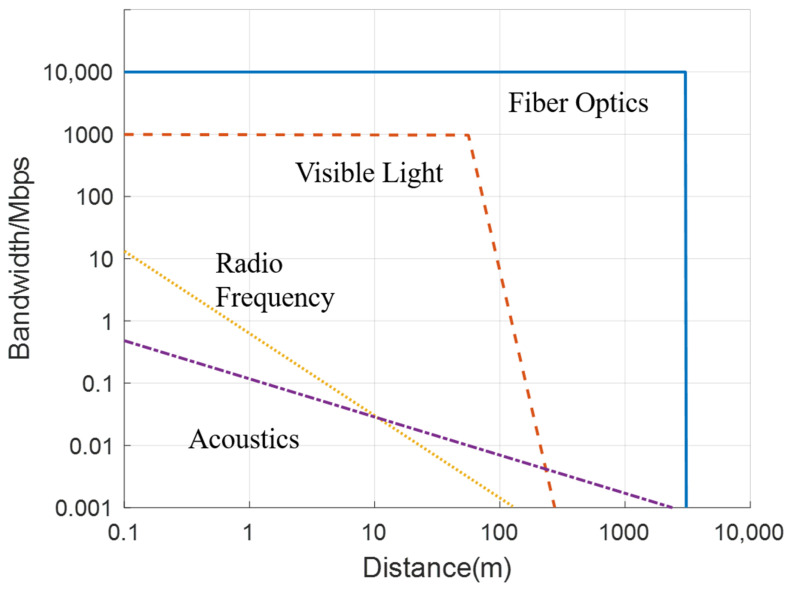
Comparison of the different underwater communication channels currently available [71].

**Figure 10 sensors-21-07849-f010:**
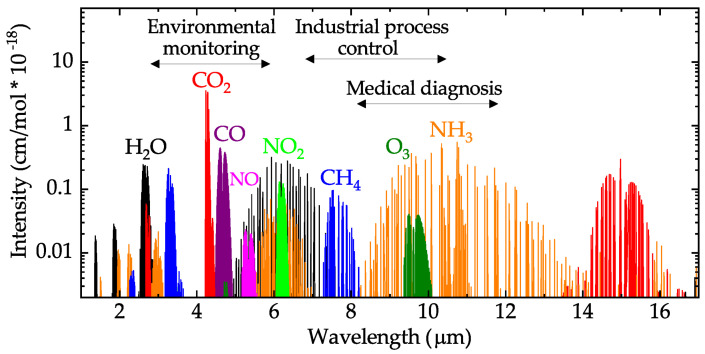
Mid-infrared absorption spectra of some gases [92].

**Figure 11 sensors-21-07849-f011:**
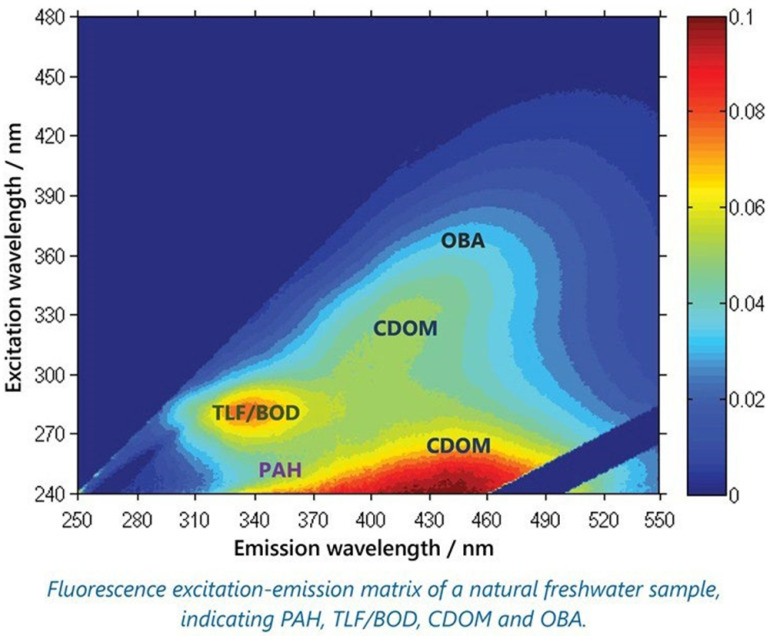
The EEM image for different OMs. Polycyclic aromatic hydrocarbons (PAH), BTEX, CDOM, tryptophan-like fluorescence (TLF), BOD, and optical brightening agents (OBA) [99].

**Figure 12 sensors-21-07849-f012:**
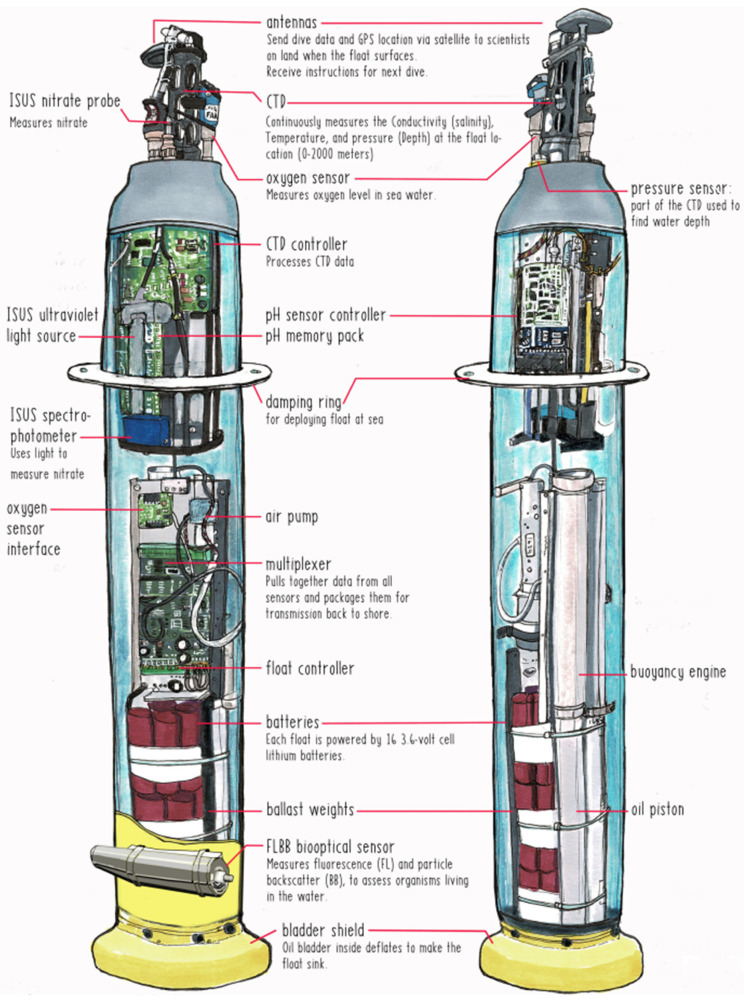
Schematic of a sort of Argo Float [104].

**Figure 13 sensors-21-07849-f013:**
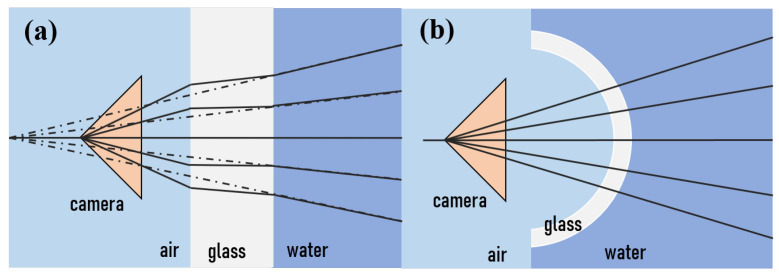
(**a**). Imaging distortion effect of non-domed shape viewport. (**b**). Domed shape viewport will release the imaging distortion effect [134].

**Figure 14 sensors-21-07849-f014:**
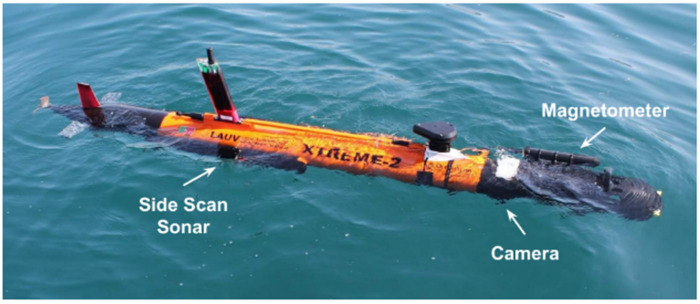
LAUV-Xtreme-2 AUV integrated with Side-scan Sonar, Underwater Camera and Magnetometer [140].

**Figure 15 sensors-21-07849-f015:**
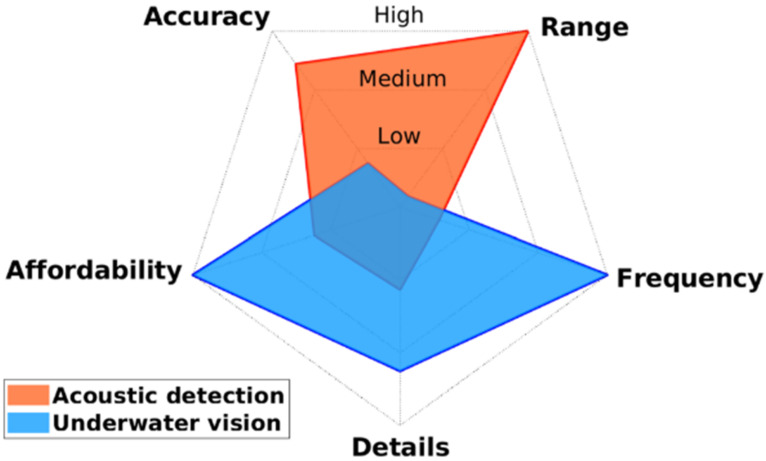
Comparison between acoustic detection and underwater vision [145].

**Figure 16 sensors-21-07849-f016:**
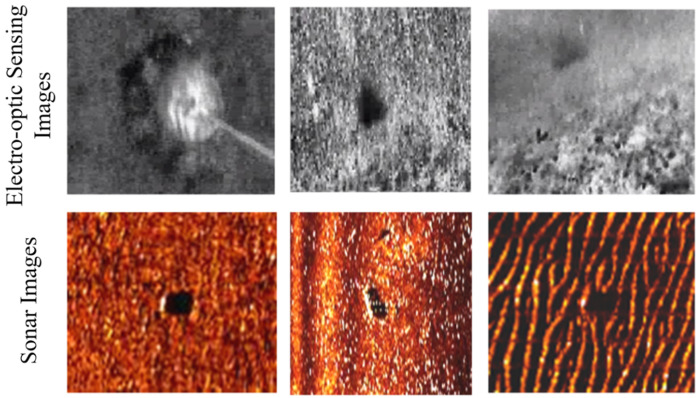
Mine images with variable environmental noises from electro-optic sensors and sonars [146].

**Figure 17 sensors-21-07849-f017:**
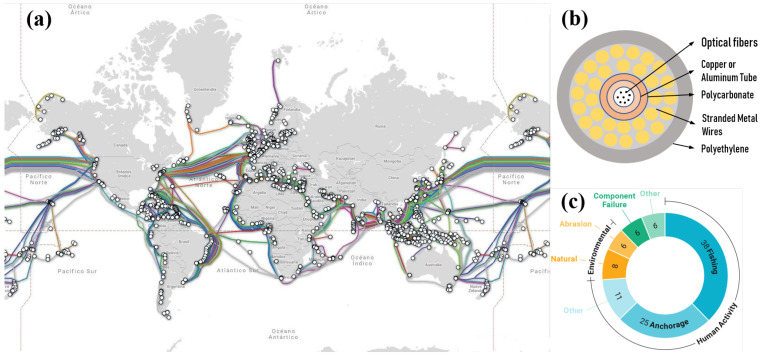
(**a**) Submarine cables around the world [152]. (**b**) Cross section of submarine communication cable [153]. (**c**) Causes of faults for cables [150].

**Table 1 sensors-21-07849-t001:** Comparison of single beam, side scan, and multi-beam sonar [22,23].

	Single-Beam Sonar	Side-Scan Sonar	Multi-Beam Sonar
Number of Beams	1	2	256 (typical)
Number of transducers	1	2	1
Coverage	Solid angle size of the beam	Two scan beams tilted away from the vessel, up to 240°	Up to 160° directly below the vessel; Theoretically up to 320° with a Dual Head MBES
Deployment Position	Bottom of vessels or subs	Sides Bottom of vessels or subs	Bottom of vessels or subs
Nadir Zone Achievable	Yes	No	Yes
Ability to accurately resolve vertical features	No	No	Yes
Ability to map irregular seafloors	No	Fair	Excellent

**Table 2 sensors-21-07849-t002:** Three main categories of underwater navigations [56].

Classifications	Principles	Methods	Characteristics
Inertial/dead reckoning	uses accelerometers and gyroscopes to estimate the current state	Magnetic compass, barometer or pressure sensor, DVL, INS	Increasing and unbounded position error
Acoustic Navigation	measuring the time of flight (TOF) of signals from acoustic beacons to perform navigate	LBL, UBL, USBL	Depending on beacons
Geophysical Navigation	use external environmental information as references for navigation	Magnetic field maps, visual-based seabed images, identify feature acoustically	Depending on sensors to identify environmental features

**Table 3 sensors-21-07849-t003:** Advantages and disadvantages of pressure sensors used for CTDs.

Pressure Sensors	Advantages	Disadvantages
Piezoresistive	simple structure, small size, high precision	low robustness
Capacitive	simple structure, high precision, high robustness	large non-linear error
Resonant	stable construction, high precision, high stability	complex manufacturing and high cost

**Table 4 sensors-21-07849-t004:** Commercially used marine environmental parameters and their applications.

Category	Objectives	Sensor Type	Working Principle	Calculation Theory	Representative Sensors	Reference
Physical EOVs	CTD:Depth	Pressure-sensitive	The external pressure results in the change of electric-signals which can reflect the ambient water pressure.	Liquid Pressure Formula	SBE 41/41CP Argo CTD; Rockland Scientific MicroCTD; OTT CTD Sensor	-
	CTD:Temperature	Thermistor	The resistance change of thermistor is related to the change of temperature.	Steinhart-Hart Equation		[105]
	CTD:Salinity	Electric	Water salinity is proportional to its conductivity.	Empirical Formula		[106,107]
Biochemical EOVs	Turbidity	Optical IR	Certain frequency IR pulse is transmitted to water, and the intensity of scattered or passed IR light detected	Empirical Formulas	Aanderaa Turbidity Sensor 4112; Chelsea Technologies UniLux Turbidity	[108,109]
	Dissolved Oxygen	Optical Blue Light (Reagent Needed)	DO pass through semi-permeable membrane and reacts with substrate film attached to fluorescence-sensitive substances. Blue light excites the fluorescence quenching reaction	Stern-Volmer Equation	SBE 63 Optical Dissolved Oxygen Sensor	[110,111,112]
		Electrochemical	DO involved redox reaction generates an electrical current, whose amplitude is directly related to the DO concentration.	Redox Reaction Electrochemical Equations	SBE 43 Dissolved Oxygen Sensor	[110,111]
	Dissolved CO_2_	Optical IR	Dissolved CO_2_ passes through silicone membrane into a detection chamber. The absorption of IR intensity is proportional to the concentration of the CO_2_	Lambert-Beer Law	SubCTech Underwater CO_2_ Sensor MK5	
	pH	Electrochemical	Electro-potential caused by different $H^{+}$ concentration between the reference and analyte solution.	Nernst Equation	SBE SeaFET V2 Ocean pH Sensor; Seanic pH probe	[113,114]
	Dissolved Organic Matter	Optical UV	High energy UV excite the fluorescence of different DOMs.	Excitation-Emission Matrix (EEM)	SBE HydroCAT-EP	[97,102]
	Nutrients: Nitrate	Optical UV	Concentration derivates from the intensity difference of the incident and transimission lights.	Lambert-Beer Law	SBE SUNA V2	[115,116,117]
	Nutrients: Phosphates	Optical Visible—IR (Reagent Needed)			SBE HydroCycle-PO_4_	
	Nutrients: Ammonium	Electrochemical Potentiometric	The open-circuit potential (OCP) between the working and reference electrodes will be measured by a high impedance voltmeter without current flow. Since only the target ions can pass the membrane on working electrode, the OCP can reflect the concentration of target ions	Concentration of ions is relevant to the OCP	Xylem ISE sensor for ammonium-WTW	[118,119]

## Data Availability

Data sharing not applicable.

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
