# Peer review of "Review of Underwater Sensing Technologies and Applications"

_sensors, 2021, doi:10.3390/s21237849_

Round 1

Reviewer 1 Report

This manuscript contains an extensive state-of-the-art survey of underwater sensors and their applications that have been reported in the literature. The paper is likely to be useful to future researchers. However, the manuscript should be carefully edited, preferably by professionals, for grammatical correctness. There exist numerous grammatical errors and many sentences, especially in the introductory part of the paper, need to be rephrased so that the reader can comprehend the intended message.

The following recent publication could also be cited and discussed in the revised manuscript:

Underwater robot sensing technology: a survey by Yang Cong et al. in vol. 1, Issue 3, Fundamental Research, ScienceDirect, pp. 337-345, May 2021. The contents of this paper and the sensors (but not the applications) part of the manuscript are somewhat similar.

Author Response

see the attachted file of reply to reviewer 1. 

Reviewer 2 Report

The authors provide a review of ocean sensing technologies, including geological exploration, navigation and communication, environmental parameters, object detection.

The paper is too broad (and superficial) for a comprehensive review of all the fields of application. Each of the main sections could be expanded and deepened to form a paper by itself, while in this form many references are missing. In the current form, it looks like a lay magazine or dissemination article. It is suggested to focus on a field of underwater technologies or to split the paper in several surveys. Moreover, some information is wrong, e.g. when it is claimed (with no reference) that OFDM is “the most suitable modulation method for underwater acoustic communication” see for example [1] below.

Minor comments:

  • Add citations in page 3 (line 107 and following)
  • What reported in line 120 (and following), is true for all types of sonars, not only side-scan
  • Add citations in line 184 and following
  • Add citations in line 205-206
  • Line 231 seems like an advertisement
  • Table 2 reports only acoustic-based positioning. It would be nice to include other techniques too
  • Introduce acronyms like VRU, CTD, etc. when using them the first time
  • Add citations in line 303 and following
  • Add citations in line 346 and following

[1] S. Mangione, G. E. Galioto, D. Croce, I. Tinnirello and C. Petrioli, "A Channel-Aware Adaptive Modem for Underwater Acoustic Communications," in IEEE Access, vol. 9, pp. 76340-76353, 2021, doi: 10.1109/ACCESS.2021.3082766.

Author Response

see the attached file of reply to reviewer 2. 

Reviewer 3 Report

Review of Underwater Sensing Technologies and Applications

Kai Sun et al.

Submitted to Sensors

14 October 2021

Summary

This paper presents Underwater Sensing Technologies and Applications with an appropriate compromise between breadth and depth, but more on the breadth side.  Applications are more applied rather than science driven, with focus on mapping, resource exploitation, environmental monitoring and security. One could up front say this review is biased toward the authors’ interests, and can’t be all inclusive, which is fine.

There is some mismatch between sensor technology - the title – and what I call infrastructure elements such as auvs, auv docking and cabled nodes, and services like power/energy, communications, and navigation, or more generally PNT – positioning, navigation, timing. One can have all the wonderful sensing tech, but if the supporting infrastructure elements and services are absent, then all for naught. The paper needs to distinguish these aspects more clearly.

I recommend revision to address these points, and the detailed comments below.

The English is passable, but a thorough style, grammar and technical editing is necessary by a native speaker, to make it more precise.

Details

Sonar does not emit the next ping until the previous one has been received, otherwise 98

the echo signals will be confused.

Not correct – can have multiple pings in the water at the same time

As the speed of sound in water is almost a con- 99

stant, the slower of the sonar carriers the denser sonar scans.

Don’t understand

The seafloor investigation and mapping have great importance on the research of 137

the marine geology, navigation safety and resources exploration.

Would seem this paragraph deserves a new sub heading 2.x level– no longer sub bottom profiler.

This is a big transition from previous sections, bringing in cable systems, seismometry, disaster warning, etc.  move later. Expand each topic?

Comparing with sonars, the air-borne sensing devices are more efficient and conve- 148

nient but with relatively lower resolution.

I would mention altimetry first, as most directly following early, deeper mapping, followed by shallow lidar etc.

By sending seismic waves to seafloor and analyzing the intensity and travel 205

time of the return seismic waves, features of sublayers underwater can be characterized.

Reference above figure

Currently, AUVs can navigate flexibly and autonomously un- 218

derwater for long term work, relying mainly on inertial navigation, acoustic navigation, 219

and some other auxiliary methods including geophysical fields, underwater imaging, 220

and seafloor topographic libraries[44].

Make clearer that AUVs are still very limited in range and duration and almost always tied to a surface support vessel. What is needed, and only just now starting are seafloor/underwater docking stations (large batteries, fuel cells, or cabled) to have “resident” AUVs that can stay down and work quasi continuously.

They are also limited by limited acoustic navigation ranges.

 I suggest looking at:

Howe, B. M., J. Miksis-Olds, E. Rehm, H. Sagen, P. F. Worcester, and G. Haralabus, Observing the Oceans Acoustically, Frontiers of Marine Science, OceanObs’19 Special Issue, doi: 10.3389/fmars.2019.00426, 2019.

This suggests a multi-purpose acoustic network covering multiple scales from small to basin scale.

Fiber optic communication enables both long distance communication and high- 281

speed rate transmission at the same time. It is the best way to be used for underwater 282

communication. But the disadvantage of fiber optic communication is also obvious, it 283

must make a physical connection between transmitters and receivers. If the cable is 284

thick, it will greatly influence the maneuverability of the submersible and if the cable is 285

thinner it can be easily broken.

Make clearer distinction between cabled observatory/node systems supporting acomms and ac nav, from fiber tethers to mobile platforms.

sound speed is accelerated by 1.4m/s. – wrong science wording! Speed is not accelerated!  Increased…

above paper references 700 km acoustic positioning (+/- 100 m), and 8,000 km acomms (11 bps).  Could include those for longer range values.

The future underwater communication will still be dominated by fiber optics 354

for long distance communications. Base stations connected by fiber optics will be laid in 355

the ocean, and information will be exchanged remotely between submersibles through 356

the base stations.

358 4. Marine Environmental Parameters

Human activities in coastal areas are increasing, because the exploration and ex- 359

ploitation of the ocean are becoming

Mention Essential Ocean Variables

https://www.goosocean.org/index.php?option=com_content&view=article&id=14&Itemid=114

of 1978 (PSS-78)[66] has become the mostly used and accepted salinity measurement 387

method,

  1. use most recent equation of state of sea water

http://www.teos-10.org/

equations 1-4 can be omitted – too much detail out of place

need to emphasize more “oceanography” as compared to environmental/pollution/mining.

503 4.7. Methane

Without

Should mention widely distributed methane clathrates. And risk to climate change – very strong green house gas, risk of massive release of seafloor becomes unstable/temperature rising.

Plastics – different from other transient pollutants – it remains and accumulates in the ocean, just breaks down in size.  There is now as much plastic mass in the ocean as biomass – and still growing!         

By now, more than 3 million kilometers of pipelines and 8,000 kilometers of cables 668

have been laid for energy transmission and worldwide telecommunications on the 669

seabed [6]. 670

check these fibres.  There are about 1.4 million km of in-service telecom cables

Fig18 -should show a standard long distance repeatered system submarine telecom cable – typically 17 mm, no armor, 8 fiber pairs in 1 mm tube in center.

As demand grows, AUVs and HROVs are considered as the 678

next generation of the best alternatives for subsea inspection.

HROV – hybrid?

Need to say that supporting infrastructure elements and services are needed – power/energy foremost, then navigation and comms.  Otherwise “swarms” will be useless; comms touched on here.

Groups of AUVs or schools of robotic fish for ocean exploration will be developed 725

and applied more and more. Due to the serious delay of underwater communications, it 726

is urgent for groups of AUVs to realize intelligent path planning and cooperation among 727

each other.

So not just comms, power and PNT.

Author Response

see the reply to reviewer 3. 

Round 2

Reviewer 1 Report

The revised paper adequately addresses the main concerns expressed by the reviewer.

Author Response

The reviewer 1 did not raise any question so we do not make specifif reply to this reviewer here. 

Reviewer 2 Report

The authors have addressed most of the comments and added several citations. However, few other steps should be considered before publication:

  • Line 203, what is METS? Please explain end introduce acronym
  • In Table 2, DVL is used before acronym is introduced
  • In line 271, please add reference
  • Add citations in lines 280-288
  • Table 4 is too small for reading. Please increase font size!
